# The Role of Chondrocyte Hypertrophy and Senescence in Osteoarthritis Initiation and Progression

**DOI:** 10.3390/ijms21072358

**Published:** 2020-03-29

**Authors:** Yeri Alice Rim, Yoojun Nam, Ji Hyeon Ju

**Affiliations:** 1Catholic iPSC Research Center, College of Medicine, The Catholic University of Korea, Seoul 06591, Korea; llyerill0114@gmail.com (Y.A.R.); givingtreemax@gmail.com (Y.N.); 2Division of Rheumatology, Department of Internal Medicine, Seoul St. Mary’s Hospital, Institute of Medical Science, College of Medicine, The Catholic University of Korea, Seoul 06591, Korea

**Keywords:** osteoarthritis, articular cartilage, chondrocyte, hypertrophy, senescence

## Abstract

Osteoarthritis (OA) is the most common joint disease that causes pain and disability in the adult population. OA is primarily caused by trauma induced by an external force or by age-related cartilage damage. Chondrocyte hypertrophy or chondrocyte senescence is thought to play a role in the initiation and progression of OA. Although chondrocyte hypertrophy and cell death are both crucial steps during the natural process of endochondral bone formation, the abnormal activation of these two processes after injury or during aging seems to accelerate the progression of OA. However, the exact mechanisms of OA progression and these two processes remain poorly understood. Chondrocyte senescence and hypertrophy during OA share various markers and processes. In this study, we reviewed the changes that occur during chondrocyte hypertrophy or senescence in OA and the attempts that were made to regulate them. Regulation of hypertrophic or senescent chondrocytes might be a potential therapeutic target to slow down or stop OA progression; thus, a better understanding of the processes is required for management.

## 1. Introduction

The simple yet complicated architecture of the articular cartilage compromises its natural capacity for self-repair [1]. The cartilage, with an avascular and alymphatic nature, is composed of dense layers of extracellular matrix (ECM) containing embedded distributed chondrocytes [2]. Cartilage lines the surface of the bone and provides a low-friction surface by absorbing external pressure or stimuli, which enables painless joint movement [3]. Osteoarthritis (OA) is the most common joint disorder throughout the whole human population. The incidence of OA is expected to increase due to the aging population throughout the world. OA accompanies progressive degradation of the articular cartilage, which leads to a loss of joint mobility and function and eventually to a low quality of life in patients due to both pain and restricted lifestyles. OA is characterized by pain, joint stiffness, and joint swelling. Severe stages of OA also include symptoms such as osteophyte and synovitis formation, which are outgrowths of bone or synovial membrane followed by inflammation. ECM degradation caused by matrix-degrading enzymes, including aggrecanases (a disintegrin and metalloprotease with thrombospondin motifs (ADAMTSs)) and matrix metalloproteinases (MMPs), is a well-known characteristic of OA. Healthy chondrocytes usually display moderate metabolic activity and proliferation under normal conditions; however, some articular chondrocytes lose their differentiated phenotype under diseased conditions and enter an endochondral ossification (EO)-like state of proliferation along with abnormal hypertrophic differentiation [4,5]. Cellular senescence can also occur alongside hypertrophy due to similar stimuli. Cellular senescence and hypertrophy share various markers and processes, and both events are reported to play a role in the development of OA. This review focuses on the hypertrophic changes and cellular senescence in chondrocytes under OA conditions.

## 2. Nature of Articular Cartilage

Articular cartilage mainly refers to hyaline cartilage. Hyaline cartilage is a smooth, lubricated surface of the cartilage that is 2 to 4 mm thick [6]. It is composed of a dense layer of ECM that also contains chondrocytes and water. Chondrocytes are quiescent cells, yet they actively synthesize a large volume of ECM proteins such as collagen, hyaluronans or glycoproteins, and proteoglycans [7,8,9]. Because of the dense layer of ECM between each lacuna, which is the small space containing each chondrocyte in the cartilage, chondrocyte only occupies 1%–5% of the total volume of the cartilage tissue. Cartilage is an avascular and alymphatic tissue, and therefore chondrocytes rely on the diffusion of nutrients from the articular surface [10]. Given these environmental characteristics, the inner layers of cartilage maintain a low oxygen environment with a low turnover rate of chondrocytes.

SRY-box 9 (*SOX9*) is the major transcription factor that regulates chondrogenesis. Chondrocytes express *SOX9*, which activates ECM genes such as collagen type II, alpha 1 (*COL2A1*), and aggrecan (ACAN). Other ECM genes, including collagen type IX and XI, were also shown to be regulated by *SOX9* [11]. Collagens and other ECM molecules, especially hyaluronan, are water-retentive, and are responsible for the high water content of cartilage. Up to 80% of the wet weight of cartilage consists of water. Collagens form about 60% of the dry weight of the cartilage, which makes them the most abundant type of protein found in ECM [9]. Although the superficial layers of cartilage mostly consist of collagen type II, the terminally differentiated hypertrophic chondrocytes in the deep zone actively synthesize collagen type X.

## 3. OA and Chondrocyte Hypertrophy

Chondrocyte hypertrophy and cell death are natural phenomena that usually occur during a developmental process called EO. Hypertrophic chondrocytes appear and play a crucial role in EO. Hyaline cartilage can be divided into two groups, (1) temporary and (2) permanent cartilage. Healthy cartilage is usually called permanent cartilage or resting chondrocytes, which are present in the articulating joint. Usually, permanent cartilage has a low proliferation rate and does not undergo terminal differentiation and EO under normal conditions [12]. Temporary cartilage is initially formed as cartilage, but the final product is bone. Unrestricted differentiation of precursor cells into the chondrocyte lineage does not lead to permanent cartilage but instead leads to bone [12]. Chondrocytes undergo active proliferation and generate a cascade of cells; whereas some of them undergo enlargement, others undergo hypertrophical changes and become hypertrophic chondrocytes. These cells increase their volume dramatically and the surroundings become mineralized to develop bone tissue [13]. The elastic nature of cartilage begins to change and harden through calcification. This makes it more difficult for the chondrocytes to receive nutrients, as most of the cells undergo apoptosis and leave small cavities within the tissue, which leaves enough room in the hardened bone for blood vessel invasion. Through this process, the cartilage turns into trabecular bone. However, the major highlight events of EO, such as chondrocyte proliferation, hypertrophic differentiation of chondrocytes, cell death, calcification or mineralization, blood vessel invasion, and chondrocyte apoptosis, occur equally in OA (Figure 1).

Cell hypertrophy generally refers to an increase in cell size and volume. Hypertrophic differentiation of chondrocytes can also be characterized by the high expression of collagen type X, runt-related transcription factor 2 (*RUNX2*), and MMP13. Hyaline cartilage markers, such as aggrecan, collagen type II, and *SOX9*, are decreased in the hypertrophic cells. Although hypertrophical changes in chondrocytes are required in bone growth and development, the mechanism acts as a double-edged sword in diseased environments [14].

Hypertrophic differentiation is also an important pathological change in OA cartilage [15]. Cartilage tissue fractures at sites that receive high mechanical stress, which is a process called ‘fibrillation’ [3]. During fibrillation in the cartilage, the tissue loses several ECM proteins, including proteoglycans and collagen fibers. Abnormal chondrocyte hypertrophy in articular cartilage is seen in cartilage destruction, which is a potential therapeutic target for OA treatment [16,17]. Hypertrophic chondrocytes start to express genes related to osteogenic differentiation and start to produce mineralized ECM proteins [18,19]. It is unclear if the abnormally differentiated hypertrophic chondrocytes give rise to fully differentiated bone cells; however, apoptosis of hypertrophic chondrocytes has been suggested in previous studies. Lacunar emptying within OA cartilage has been observed, which led to the loss of the articular cartilage [20,21]. After these changes, the subchondral bone generates osteophytes. Chondrocyte calcification by calcium deposition is the final stage of chondrocyte hypertrophy [12]. Previous studies report that the layer bordering the calcified cartilage and the non-calcified cartilage increases after the age of 60 years [22]. Although this layer of calcified cartilage tends to increase throughout the aging process, in some cases, the calcified region decreases instead. This indicates that the lower calcified cartilage is replaced by bone due to the EO-like process in OA.

## 4. OA and Chondrocyte Senescence

OA and cellular senescence are characterized by various epigenetic changes due to altered cellular phenotype and disease progression. Therefore, cellular senescence is also a factor that is thought to play a significant role in the pathology of OA [23]. Cellular senescence can be caused by various factors, including external stimuli or stress, oncogene activation, radiation, oxidative stress, telomere length reduction, and abnormal DNA replication. OA chondrocytes exhibit features that are similar to senescent cells, such as telomere shortening and increased senescence-associated β-galactosidase (SA-βgal) activity. The term ‘cellular senescence’ was first mentioned in the 1960s by Leonard Hayflick and Paul Moorhead, after finding that in vitro cultured primary human fibroblasts have a restricted lifespan, which is approximately 50 cell divisions [24]. Senescent cells undergo morphological changes such as flattening due to increased cell size, vacuolization, and accumulation of stress granules. Growth arrest, resistance to cell death, and altered gene expression that leads to the production of bioactive senescence-associated secretory phenotype (SASP) are induced due to these morphological changes [25,26]. SASPs are various inflammatory cytokines and growth factors secreted by senescent cells. Interestingly, SASPs that are reported to be produced by senescent cells (i.e., GM-CSF, IL-1α and β, IL-6, IL-7, IL-8, MCP-1, MCP-2, MMP1, MMP3, MMP10, MMP13, TIMP-2, and more) have been found in OA tissues or in synovial fluids [27]. SASPs are produced by senescent cells that secrete various inflammatory cytokines and growth factors [6,28]. The growth arrest of senescent cells is caused by the expression of cell cycle inhibitors that can be induced by various stimuli such as DNA damage, telomere shortening, and oncogene activation [25]. These genetic changes can be induced by other stimuli such as reactive oxygen species (ROS), mitotic stress, and inflammation. When chondrocytes are isolated from native culture and maintained in an in vitro culture as a monolayer, they easily lose their characteristics and express various senescence and dedifferentiation-related genes [29]. Senescence and dedifferentiation of chondrocytes during culture is accompanied by decreased expression of chondrocyte-specific proteins such as collagen type II, and other glycoproteins [30].

Although various cell types are involved in OA pathology, chondrocytes are primarily thought to play a major role in OA induction by cellular senescence [23]. Two different mechanisms of senescence are suggested in chondrocytes: (1) replicative senescence and (2) stress-induced premature senescence (SIPS) [31]. There are two main aging pathways, where p53/p21 is thought to act on replicative senescence as well as chondrocyte apoptosis, and p38/p16 induces SIPS [31,32]. Replicative senescence generally occurs after 30–40 passaging and shows features of senescent phenotype including enlarged flattened cells in culture and the expression of SA-βgal. Chondrocytes have also been shown to have telomere shortening with age [33]. Because of this, chondrocyte senescence is thought to be closely related to OA, which also occurs with age. Given the chondrocyte characteristics, it is unlikely that we will find evidence of cell division in normal chondrocytes. However, chondrocyte proliferation is a feature observed in OA as well as telomere shortening and SA-βgal expression [34]. Although replicative senescence is caused by telomere shortening, SIPS is caused by oxidative stress and DNA damage without the change in telomere length [35]. Cellular senescence can be caused by chronic stress in cells; however, there is still the possibility that post-traumatic OA can be characterized or triggered by the senescent cells within the damaged region [23]. In the tissue of one patient, however, senescent cells were found to be absent in normal cartilage and instead were observed near the osteoarthritic lesions [36]. When senescent cells were transplanted into the knee joint of wild type mice, an OA-like state was induced, which included pain, impaired mobility, and morphological and histological changes [37]. SASPs secreted by senescent cells can alter the tissue microenvironment and impair tissue regeneration induced by stem cells or progenitor cells, which can eventually lead to the senescence of the neighboring cells [38].

Beside chondrocytes, synovial fibroblasts are also thought to initiate or progress OA through senescence. Nuclear expression of p16 was detected in higher amounts in OA synovial tissue samples when compared to that of normal synovial tissues, which indicates senescence in OA synovial fibroblasts [39]. Senescent synovial fibroblasts induced by H_2_O_2_ or tumor necrosis factor-α (TNFα) expressed increased CDKN1A and CDGN2A and other pro-inflammatory SASP-associated factors such as IL-6, CXCL8, CCL2, and MMP3.

## 5. Hypertrophy and Senescence-Related Markers in OA

Several markers are known as chondrocyte hypertrophy and senescence markers. Both hypertrophy and senescence are not usually characterized by a specific set of markers, but are rather associated with a set of cellular phenotypes that often co-exist in a stressed environment (Figure 2) [23]. Although both chondrocyte hypertrophy and senescence are not clearly understood, markers such as IL-1β, collagen type X, *RUNX2*, vascular endothelial growth factor (VEGF), osteopontin, osteocalcin, and Indian hedgehog (IHH) are thought to be related [31,40]. The accumulation of MMPs induced by pro-inflammatory cytokines is also a positive senescence marker in OA chondrocytes.

Collagen type X is a major marker used to detect chondrocyte hypertrophy. This type of collagen is usually not expressed in healthy articular cartilage, though it is thought to play a role in the early stage of endochondral bone formation since it has been detected at sites of hypertrophic chondrocyte regions and calcification. Protein and mRNA levels of collagen type X are also detected in human OA cartilage but their expression shows significant local variation [12,16,41,42,43]. Despite the overall expression of collagen type X, alpha 1 (*COL10A1*) was increased in human OA cartilage, and the expression was higher in the relatively less degenerated area [44]. Isolated chondrocytes cultured in vitro lose their normal morphology and start synthesizing collagen type X after several passages [29,45]. For example, whereas passage 2 chondrocytes show high levels of collagen type II and low levels of collagen type X, passage 6 chondrocytes show the opposite characteristics with increased collagen type X [31]. The exact mechanism of senescence and collagen type X is not fully understood, but its expression is thought to be the main reason for chondrocyte dedifferentiation [46].

Elevated levels of MMP13 have been reported in OA cartilage [47,48]. The gene level of MMP13 was highly expressed in OA cartilage when compared to normal levels. Elevated expression of MMP13 and collagen type X was reported after inducing knee joint stability by the transection of knee ligaments and meniscus removal in mouse OA models [49]. MMP13 is also thought to play a central role in the irreversible degradation of collagen type II in OA [50,51]. Knockdown of activin-like kinase 5 (ALK5) was reported to increase MMP13 mRNA levels [52]. TGFβ is important for the maintenance and protection of healthy articular cartilage; therefore, it is commonly used in the in vitro culture of chondrocytes or chondrogenesis. The binding of TGFβ to its receptor recruits the type I receptor ALK5. In turn, this molecule complex phosphorylates the smad2 and 3 in the smad2/3 pathway, which is known to repress chondrocyte terminal differentiation [53,54]. Another alternative receptor ALK1 activates the smad1/5/8 signaling pathway, which is known to induce chondrocyte terminal differentiation [55,56]. The change in the ALK1/ALK5 ratio during aging or OA development may result in increased dominance of the smad1/5/8 signaling in chondrocytes and induce a hypertrophic-like state by high levels of MMP13. MMP13 is also a central regulator of chondrocyte senescence [57]. Chondrocytes undergoing senescence produce ECM-degrading enzymes such as MMP13, which promotes the degradation of the main protein of the articular cartilage, aggrecan, and collagen type II. MMP13 produced by cultured chondrocytes isolated from older donors appeared to be increased [58]. Changes in chondrocyte function (e.g., decreased anabolic activity and increased catabolic activity) are thought to be related to MMP13 production.

*RUNX2* is the main transcription factor that is involved in hypertrophic chondrocyte differentiation and early osteogenesis [48,59]. One of the hallmarks of OA is the upregulation of *RUNX2*. Therefore, *RUNX2* is assumed to be a major transcriptional factor that directly regulates the expression of matrix degradation enzymes in the damaged articular cartilage [60]. When the destabilization of the medial meniscus (DMM) osteoarthritis model was induced in *RUNX2* knockout mice, the gene expression of matrix degradation enzymes (i.e., MMP9, MMP13, ADAMTS4, ADAMTS5, ADAMTS7, and ADAMTS12) was significantly reduced compared with DMM-induced Cre-negative control. The deletion of *RUNX2* in DMM-induced mice decreased MMP13 protein levels in the articular cartilage. Cells expressing ectopic *RUNX2* showed a senescent-like phenotype that was characterized by an enlarged and flattened morphology and β-galactosidase staining; p53 signaling was required for this process [61].

A characteristic feature of hypertrophy and OA cartilage is the increased production of VEGF. VEGF induces the migration of endothelial cells by chemotactic actions and induces angiogenesis in vivo. VEGF also promotes angiogenesis in the cartilage tissue, which is related to the calcification of chondrocytes that can lead to dysregulated osteogenesis of the normal cartilage. Neoangiogenesis in the cartilage growth plate plays an important role in EO; therefore, VEGF is thought of as a critical mediator during EO. Carlevaro et al. investigated the expression of VEGF in mammalian and avian embryo long bone growth plates [62]. Although VEGF was observed in fully mature hypertrophic chondrocytes, it was completely absent in proliferating and quiescent cells in both chicken and mice. VEGF mRNA generates five different isoforms with a different number of amino acid residues by alternative splicing, labeled VEGF121, VEGF145, VEGF165, VEGF189, and VEGF206 [63]. Although only three types (VEGF121, VEGF165, and VEGF189) were detectable in both OA and normal cartilage, their receptors were only found in OA cartilage. OA cartilage secreted a significantly higher amount of VEGF than normal cartilage when cultured in vitro. VEGF expression is induced by mechanical forces on articular cartilage [64]. Therefore, it is thought that VEGF may regulate chondrocyte differentiation in response to mechanical forces [65]. After inducing DMM in mice, immunostaining showed increased levels of VEGF in articular cartilage, subchondral bone, synovium, and meniscus. 

IHH is also suggested as a marker for chondrocyte hypertrophy. IHH is a key signaling molecule that is expressed in pre-hypertrophic chondrocytes during growth plate development and that regulates chondrocyte hypertrophy during EO [66]. The upregulation of IHH in postnatal cartilage showed an increased chondrocyte hypertrophy and degradation [67]. Activation of IHH downstream signaling pathways results in a reduction in articular cartilage thickness and proteoglycan content levels, while inhibition of IHH signaling reversed these effects [68,69]. IHH is reported to regulate the onset of chondrocyte hypertrophy by a negative feedback loop with parathyroid hormone-related protein (PTHrP) [70]. IHH activates the expression of PTHrP in particular chondrocytes and PTHrP signals through the receptor PTHR1 to inhibit chondrocyte hypertrophy and maintain the proliferating state. Mak et al. reported that IHH signaling can also directly regulate chondrocyte hypertrophy in the absence of PTHrP. IHH also promoted chondrocyte hypertrophy through Wnt/β-catenin activation and bone morphogenetic proteins (BMPs) signaling.

Expressions of p16, p21, and p53 increase in senescent cells, which result in cell cycle arrest. Articular chondrocytes with increased expression of p16 and p21 exhibit increased senescence. Chondrocytes expressing p53 showed similar morphology to OA chondrocytes and underwent apoptosis, whereas downregulation of p53 expression prevented chondrocytes from undergoing senescence or apoptosis [71]. Chondrocytes expressing p16 lose their normal phenotype and express increased inflammatory cytokines and MMPs such as IL-1β, IL-6, and MMP13 [46]. In chondrocytes isolated from cadaveric donors without evidence of OA between 17 to 72 years old, age was found to be responsible for 27% of the variation in p16 levels. However, loss of p16 was not critical enough to prevent the development of age-induced or injury-induced traumatic OA [72]. Collagen type II expression is negatively correlated with p21 expression [73].

Oxidative stress caused by ROS is a major regulator in cell senescence. ROS induce multiple genes that spark senescence or dedifferentiation in chondrocytes. The main ROS reported to induce cartilage damage in chondrocytes are hydrogen peroxide (H_2_O_2_) and peroxynitrite [74,75]. Degenerated cartilage regions in OA patients had significantly lower anti-oxidative potential than the intact cartilage region, confirming increased oxidative damage in degenerated cartilage [76]. Chondrocytes treated with H_2_O_2_ showed shorter telomere length in vitro. When OA cartilage tissue was cultured with H_2_O_2_ addition, the number of glycosaminoglycans (GAGs) gradually decreased in a time-dependent manner. However, the addition of an anti-oxidative agent reversed these effects, inhibited GAG loss, and maintained telomere length. H_2_O_2_ addition significantly increased p21 expression in both early and late passage chondrocytes [77]. The expression started to increase one hour after treatment and reached its highest level after three hours, and the change in early passage chondrocytes was nine-fold higher than the initial level.

Growth factors such as BMPs and other cytokines are reported to play a role in chondrocyte hypertrophy. Several BMPs enhance chondrogenesis; however, dysregulated levels of BMPs may induce calcification of cartilage and accelerate cartilage degradation. During OA development, chondrocytes reportedly express BMPs [9,78]. The BMP canonical smad1/5/8 pathway is thought of as an inducer of chondrocyte hypertrophy and EO [79]. Cytokines such as IL-6 and IL-8 are known to induce growth arrest of senescent cells when secreted into the cell microenvironment. IL-1β is associated with chondrocyte dedifferentiation, and induces repression of *SOX9* and the synthesis of nitric oxide (NO) [31,80]. The induction of NO by IL-1β activates c-Jun N-terminal kinases (JNK) signaling, which is a well-known inducer of senescence [81]. IL-7 is a cytokine that is upregulated age-dependently [27]. The expression of IL-7 is higher in cultured articular chondrocytes from older donors compared to that of the younger donors. Furthermore, increased levels of MMP13 are induced in chondrocytes after IL-7 treatment [82].

Advanced glycation end products (AGEs) are produced from reducing sugars and free amino groups of proteins, lipids, or nucleic acids by a non-enzymatic reaction [83,84]. AGE accumulation suppresses collagen type II production and also stimulates the expression of MMPs and ADAMTSs [85,86,87]. Collagen cross-linking is increased by AGEs, which enhances tissue stiffness [88,89]. Increased AGEs are pathogenic and induce oxidative stress and inflammation. AGEs also induce the secretion of TNF-α, and NO [86,90].

## 6. Cartilage Treatment and Regeneration Strategies Targeting Hypertrophy or Senescence

Chondrocyte hypertrophy has been targeted in cartilage regeneration rather than treatment. Several strategies have been attempted to inhibit chondrogenic hypertrophy.

The addition of TGFβ into an in vitro culture was reported to enhance chondrogenesis and reduce chondrocyte proliferation and hypertrophy [91,92,93]. TGFβ1 and TGFβ3 were mostly used in in vitro chondrogenesis in various studies [94]. Some studies report that there is no significant difference between TGFβ subtypes in suppressing hypertrophy during chondrogenesis; several other studies show that TGFβ1 is superior in suppressing hypertrophy [95,96,97]. Smad3 disrupted in mutant mice showed a progressive loss of articular cartilage and symptoms that resemble OA; however, the addition of TGFβ1 reversed these symptoms, suggesting that TGFβ/smad3 signals inhibit terminal hypertrophic differentiation of chondrocytes [54]. TGFβ1 also enhanced hyaline-like properties using BMP2 in bovine synovial explants, and suppressed chondrocyte hypertrophy [98].

PTHrP is a parathyroid hormone with various isoforms, and it is involved in the development of multiple tissues including skeletal development [99]. PTHrP increased GAG contents and expression of *SOX9* and *COL2A1* in pellets generated from bone marrow stromal cells and adipose-derived stromal cells, and downregulated the expression of *COL10A1* and *RUNX2* in a dose-dependent manner [100]. Between other isoforms, PTHrP(1-34) was reported to induce chondrogenesis and reduce hypertrophy in mesenchymal stromal cells [101]. Previous work done by Lee and colleagues showed that of four PTHrP isoforms (1-34, 1-86, 7-34, and 107-139), PTHrP(1-34) showed the most significant enhancement in chondrogenesis and suppression in hypertrophy in human bone marrow stromal cells [102]. Other studies also showed that PTHrP(1-34) increased the gene and protein expression of collagen type II. Collagen type X gene expression was less affected; however, PTHrP(1-34) downregulated the expression of IHH and alkaline phosphatase (ALP) [101].

Various groups have attempted to alter the levels of MMPs despite the negative musculoskeletal side effects in humans [103]. A broad spectrum of MMP inhibitors has been proposed and tested for OA [104]. Bertram et al. confirmed the activity of proteases and MMPs in chondrogenesis and how they influenced the early and late stages of chondrogenesis [105]. The addition of pan-MMP inhibitors suppresses chondrogenesis in a dose-dependent manner, which was confirmed by decreased levels of proteoglycan disposition, collagen type II and X staining levels, ALP activity, and *SOX9* and *COL2A1* expressions. The MMP13-specific inhibitor GG86/2 did not affect early chondrogenesis and did not alter the collagen type II accumulation. GG86/2 partially reduced ALP activity. There are several drugs developed to target MMPs in OA [103]. Novartis described an active hydroxamate MMP inhibitor, CGS-27023A, which can be taken orally. This drug inhibited MMP1, MMP2, MMP3, MMP9, MMP12, and MMP13, and exhibited a chondroprotective effect in OA animal models. However, it was halted due to musculoskeletal side effects related to treatment duration during a phase I clinical trial [106]. Other companies attempted to develop MMP-13 inhibitors such as CP-54439 (Pfizer) and WAY-170523 (Wyeth Research), and were taken to phase II clinical trials. Although the trials of pan-MMP inhibitors had rather disappointing results, selective MMP13 inhibitors are under development with both improved delivery and reduced side effects [103].

Senescent cells secreting pro-inflammatory cytokines and matrix-degrading enzymes were targeted using senolytic (a combination of the words ‘senescence’ and ‘-lytic’) compounds that have emerged as a potential therapeutic [107]. Baker et al. attempted to eliminate senescent cells with a tailored drug through the expression of a transgene that is only expressed in P16(Ink4a)-positive cells [108]. The direct effect of a senolytic agent in OA was tested in an anterior cruciate ligament transection (ACLT) model induced in transgenic p16-3MR mice, which was designed to eliminate p16-positive cells with ganciclovir [28]. The senolytic compound UBX0101 was repeatedly delivered by intra-articular injections after two weeks of ACLT induction. Although the half-life of the compound is only 1.5 h, UBX0101 successfully reduced the senescent chondrocytes and the production of SASP factors and showed suppression of the progression of OA.

## 7. Conclusions

The molecular mechanisms of OA initiation and progression require considerable further study, despite significant progress in recent years. OA is mainly caused by trauma induced by an external force or cartilage damage accumulated during aging. During these processes, chondrocyte hypertrophy and senescence are thought to play a critical role in OA initiation or progression. However, the remaining question is: which came first, the chicken or the egg? There is still little understanding of whether these two independent processes (i.e., chondrocyte hypertrophy and senescence) are dependent on penetration in the other. Further study on which event is the cause or the effect should be conducted to better understand these processes. Investigating the effect of chondrocyte hypertrophy and/or senescence in OA would provide a promising research field in developing a potential therapeutic agent for OA treatment.

## Figures and Tables

**Figure 1 ijms-21-02358-f001:**
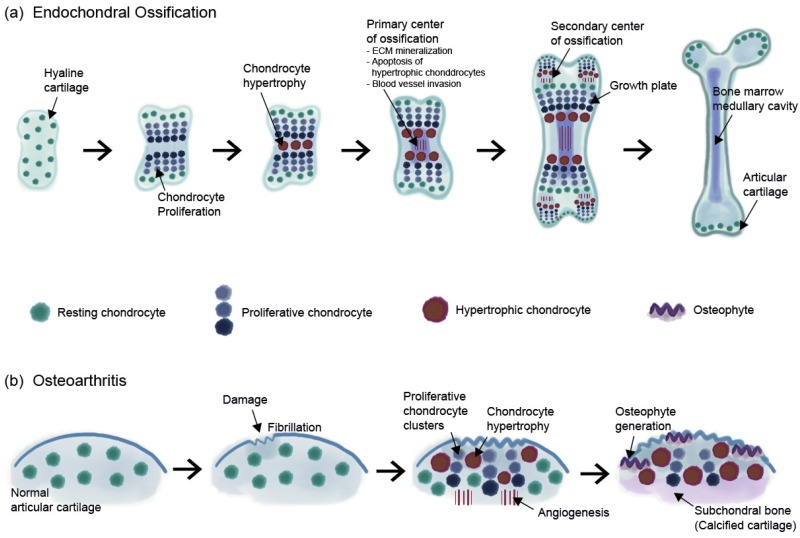
Schematic image of (**a**) endochondral ossification in the embryonic cartilage and (**b**) progression of osteoarthritis in the articular cartilage.

**Figure 2 ijms-21-02358-f002:**
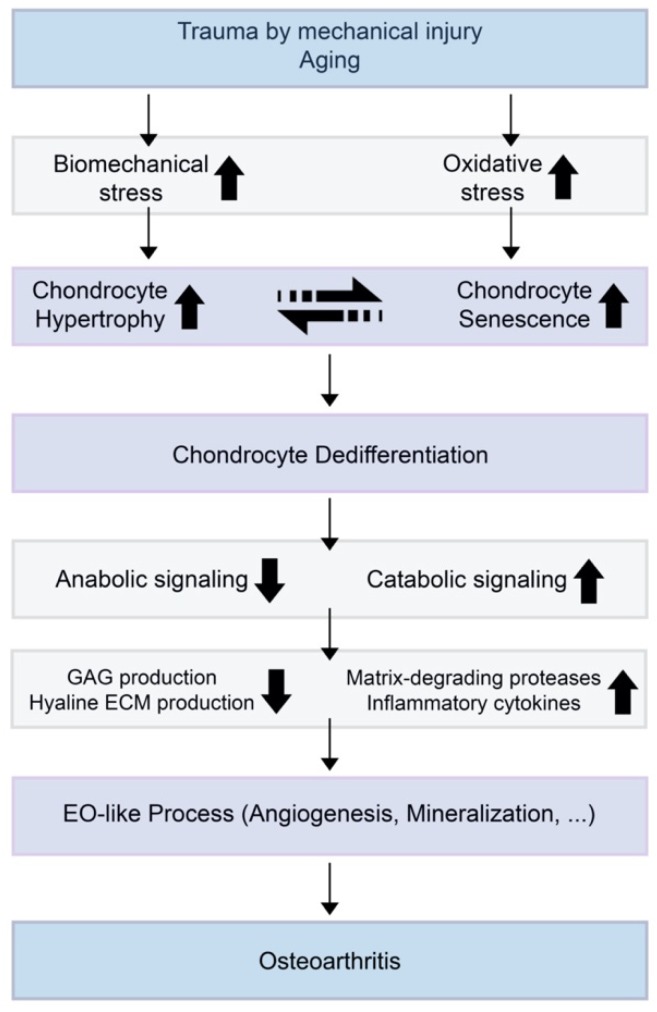
A theoretical model for the relationships of chondrocyte hypertrophy and chondrocyte senescence and the initiation or progression of osteoarthritis (OA). When trauma is induced by mechanical injury or aging, biomechanical stress and oxidative stress increase in the cartilage environment. Increased chondrocyte hypertrophy and senescence induce decreased anabolic signaling and increased catabolic signaling. This results in suppressed glycosaminoglycan (GAG) production by normal chondrocytes and increases the expression of extracellular matrix (ECM)-degrading enzymes and inflammatory cytokines.

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
