# Peer review of "The Role of Chondrocyte Hypertrophy and Senescence in Osteoarthritis Initiation and Progression"

_ijms, 2020, doi:10.3390/ijms21072358_

Round 1

Reviewer 1 Report

In this review the authors underline the role of chondrocyte hypertrophy and senescence in osteoarthritis initiation and progression. 

Osteoarthritis is a widely common disease, aged related. The exact mechanism of triggering and progression of the pathology is still not well understood. Several studies on the etiology of the pathology have been carried out and different treatments based on drugs, cell therapy based on the application of mesenchymal stem cells (MSCs) alone or in combination with biomaterials have been suggested without any success. 

Recently, the involvement of cellular senescence in the genesis and progression of several age related diseases has been suggested. Nowadays, cellular senescence is a central topic in research and the number of studies regarding cellular senescence is improving day by day. Anyway, the knowledge is still poor and more investigations are needed to exactly understand the molecular mechanism involved in the initiation of senescence and the connection with several age related diseases. For this reason, the review is extremely interesting and worthy of publication.   

This review underlines the involvement of chondrocyte senescence in OA and their similarity with hypertrophic chondrocytes, which have a pivotal role in the progression of OA. The comparison of chondrocytes senescence and hypertrophy is crucial in understanding the connection between senescence and OA. 

The review is well written and described different aspects of chondrocytes in normal and pathological conditions, underlining the common molecular markers involved in senescence and hypertrophy. Furthermore, the study is completed with a brief description of therapeutical treatments designed in counteracting cellular senescence. 

In my opinion the review is worthy of publication. Nevertheless, I have some minor doubts/suggestions for the authors

  • Although the role of chondrocytes in the OA is crucial, other cellular components such as synovial fluid stem cells and synoviocytes of the synovial membrane could have a role in the initiation and progression of the pathology. Any knowledge of senescence in synovial fluid stem cells and/or synoviocytes and their role in the pathology?

  • Pat 2, line 69: during EC, hypertrophic chondrocytes die leaving space to the new bone. You wrote that a similar mechanism happened in OA. It is not clear if in OA hypertrophic chondrocytes die and they are then substituted by bone cells, which give raise to osteophytes, or hypertrophic chondrocytes never die but start to accumulate calcified materials in the cytoplasm. Please, clarify better this point. 

  • The references are appropriate but too old. Please, update the bibliography. 

Author Response

Author’s Response to the Review Comments

  • Journal: International Journal of Molecular Sciences
  • Manuscript #: ijms-755176
  • Title of Paper: The role of chondrocyte hypertrophy and senescence in osteoarthritis

  • Authors: Yeri Alice Rim (First author); Yoojun Nam; Ji Hyeon Ju (Corresponding author)

It is with excitement that we resubmit the revised version of the manuscript ijms-755176, “The role of chondrocyte hypertrophy and senescence in osteoarthritis” for the International Journal of Molecular Sciences. Thank you for the opportunity to revise and resubmit this manuscript. We addressed all issues and concerns indicated in the review report. We appreciate the time, the efforts and the constructive suggestions by the editor and referees in reviewing this manuscript.

We have responded specifically to each suggestion below. The changes were numbered to make it easier to identify where necessary. The major changes suggested by the reviewers were highlighted or tracked in the manuscript.

Response to Comments from Reviewer 1

Osteoarthritis is a widely common disease, aged related. The exact mechanism of triggering and progression of the pathology is still not well understood. Several studies on the etiology of the pathology have been carried out and different treatments based on drugs, cell therapy based on the application of mesenchymal stem cells (MSCs) alone or in combination with biomaterials have been suggested without any success.

Recently, the involvement of cellular senescence in the genesis and progression of several age related diseases has been suggested. Nowadays, cellular senescence is a central topic in research and the number of studies regarding cellular senescence is improving day by day. Anyway, the knowledge is still poor and more investigations are needed to exactly understand the molecular mechanism involved in the initiation of senescence and the connection with several age related diseases. For this reason, the review is extremely interesting and worthy of publication.  

This review underlines the involvement of chondrocyte senescence in OA and their similarity with hypertrophic chondrocytes, which have a pivotal role in the progression of OA. The comparison of chondrocytes senescence and hypertrophy is crucial in understanding the connection between senescence and OA.

The review is well written and described different aspects of chondrocytes in normal and pathological conditions, underlining the common molecular markers involved in senescence and hypertrophy. Furthermore, the study is completed with a brief description of therapeutical treatments designed in counteracting cellular senescence.

In my opinion the review is worthy of publication. Nevertheless, I have some minor doubts/suggestions for the authors:

  1. Although the role of chondrocytes in the OA is crucial, other cellular components such as synovial fluid stem cells and synoviocytes of the synovial membrane could have a role in the initiation and progression of the pathology. Any knowledge of senescence in synovial fluid stem cells and/or synoviocytes and their role in the pathology?

Answer) Thank you for the suggestion. Following your suggestion, we have added the information on synoviocytes and senescence in part 4, line 456-461. The changes were tracked in the revised manuscript.

  1. Pat 2, line 69: during EC, hypertrophic chondrocytes die leaving space to the new bone. You wrote that a similar mechanism happened in OA. It is not clear if in OA hypertrophic chondrocytes die and they are then substituted by bone cells, which give raise to osteophytes, or hypertrophic chondrocytes never die but start to accumulate calcified materials in the cytoplasm. Please, clarify better this point.

Answer) Thank you for your kind suggestion. We understand what is confusing in that part of the manuscript. We tried to make it more clearer for the readers and changed the manuscript in part 3, line 345-350. Please check and let us know it is still confusing. The changes were tracked in the manuscript.

  1. The references are appropriate but too old. Please, update the bibliography.

Answer) Thank you for the suggestion. We tried to add and update the bibliography (within 5 years) in the manuscript. The changes and added references were tracked in the manuscript.

Reviewer 2 Report

Dear Authors,

The paper submitted was well written and documented, however i would suggest some minor points:

  1. during mechanism description i would suggest adding more images, thus 2-3 more images should be added for each point
  2. English language has some mistakes so i would suggest a thorough examination of spelling and grammar
  3. Only few of the bibliographical references are from the last 5 years, i would suggest adding some newer references

Author Response

Author’s Response to the Review Comments

  • Journal: International Journal of Molecular Sciences
  • Manuscript #: ijms-755176
  • Title of Paper: The role of chondrocyte hypertrophy and senescence in osteoarthritis

  • Authors: Yeri Alice Rim (First author); Yoojun Nam; Ji Hyeon Ju (Corresponding author)

It is with excitement that we resubmit the revised version of the manuscript ijms-755176, “The role of chondrocyte hypertrophy and senescence in osteoarthritis” for the International Journal of Molecular Sciences. Thank you for the opportunity to revise and resubmit this manuscript. We addressed all issues and concerns indicated in the review report. We appreciate the time, the efforts and the constructive suggestions by the editor and referees in reviewing this manuscript.

We have responded specifically to each suggestion below. The changes were numbered to make it easier to identify where necessary. The major changes suggested by the reviewers were highlighted or tracked in the manuscript.

Response to Comments from Reviewer 2

The paper submitted was well written and documented, however i would suggest some minor points:

  1. during mechanism description i would suggest adding more images, thus 2-3 more images should be added for each point

Answer) Thank you for the suggestion. We changed the figure based on your suggestion. Please let us know if there is more to do to improve this figureJ The edited figure was also added in the manuscript.

  1. English language has some mistakes so i would suggest a thorough examination of spelling and grammar

Answer) Thank you for the suggestion. We requested English editing through the MDPI English editing service. The changes were tracked throughout the manuscript.

  1. Only few of the bibliographical references are from the last 5 years, i would suggest adding some newer references

Answer) Thank you for the suggestion. We have tried to add some updated references (within 5 years) in the manuscript. The added references were tracked throughout the manuscript.